# Comparative Evaluation of Vitek 2 and Etest versus Broth Microdilution for Ceftazidime/Avibactam and Ceftolozane/Tazobactam Susceptibility Testing of *Enterobacterales* and *Pseudomonas aeruginosa*

**DOI:** 10.3390/antibiotics11070865

**Published:** 2022-06-27

**Authors:** Arhodoula Papadomanolaki, Maria Siopi, Polyxeni Karakosta, Sophia Vourli, Spyros Pournaras

**Affiliations:** Laboratory of Clinical Microbiology, “Attikon” University Hospital, Medical School, National and Kapodistrian University of Athens, 12462 Athens, Greece; papadomanolakia@gmail.com (A.P.); marizasiopi@hotmail.com (M.S.); p_karakosta@hotmail.com (P.K.); svourli@med.uoa.gr (S.V.)

**Keywords:** challenge isolates, carbapenem-resistant *Enterobacterales*, carbapenem-resistant *Pseudomonas*, antimicrobial susceptibility testing, BMD, major error, very major error

## Abstract

Ceftazidime/avibactam (CZA) and ceftolozane/tazobactam (C/T) are novel antibiotics with activity against multidrug-resistant Gram-negative pathogens. Nevertheless, resistance to both agents has been reported emphasizing the need for accurate and widely accessible susceptibility testing. In the present study, Vitek 2 and Etest CAZ and C/T MIC results for 100 non-repetitive clinical isolates (83 *Enterobacterales* and 17 *P. aeruginosa*, whereof 69 challenge isolates) were compared to the standard broth microdilution (BMD) method. EUCAST breakpoints were used for assessing the categorical (CA) and essential (EA) agreement between the methods along with the corresponding error rates. The Vitek 2 performance was comparable to that of BMD for testing both antimicrobial agents exceeding the ISO requirements (CA 98–99%, EA 96–100%, major errors (MEs) 0–1%, very major error (VMEs) 1%). Likewise, the Etest provided accurate results for CZA and C/T testing against *Enterobacterales* and *P. aeruginosa*, respectively (CA 100%, EA 97–100%, MEs 0%, VMEs 0%). On the contrary, EA of 85% and 6% VME rate were found for CZA Etest and *P. aeruginosa*. Overall, Vitek 2 measurements of CZA and C/T susceptibility correlated closely with the reference BMD, indicating that it can represent a suitable alternative to BMD for susceptibility testing of *Enterobacterales* and *P. aeruginosa*. The Etest did not fulfill the ISO performance criteria of EA and VME for CZA and *P. aeruginosa*. Further studies are needed to assess whether the Etest allows a reliable assessment of CZA and C/T EUCAST MICs.

## 1. Introduction

Multidrug-resistant (MDR) infections result in longer hospital stays, treatment failures, higher mortality rates and rising healthcare costs [1,2]. Recently, the World Health Organization has published a priority pathogens list for research and development of new antibiotics, among which critical-priority bacteria include third-generation cephalosporin-resistant and carbapenem-resistant *Enterobacterales* (CRE) as well as carbapenem-resistant *Pseudomonas aeruginosa* (CRPA) [3]. Meanwhile, the Centers for Disease Control and Prevention have assigned certain types of Gram-negative bacteria, such as CRE, as an “urgent threat” to human health, while extended-spectrum β-lactamase-producing *Enterobacterales* (ESBL-E) and MDR *P. aeruginosa* have been listed as a “serious threat” [4]. Worryingly, during the last few decades these bacteria tend to become resistant to almost all available antibiotics [5]. ESBL-E and CRE are responsible for a variety of community and healthcare-associated infections associated with poor clinical outcomes [6,7], whereas *P. aeruginosa* is a major nosocomial and community pathogen [1,8].

Due to the increasing emergence of ESBL- and carbapenemase-producing bacteria, there have been very limited therapeutic options against MDR Gram-negatives, among which a common strategy is the combination of a β-lactam with a β-lactamase inhibitor [9]. Ceftazidime/avibactam (CZA) is a combination of a third-generation cephalosporin and a novel non-β-lactam β-lactamase inhibitor. Avibactam is an appealing addendum to existing antimicrobial agents since it is active against all Class A and Class C β-lactamases, including ESBLs and *Klebsiella pneumoniae* carbapenemases (KPCs), as well as some class D β-lactamases, but not against Class B metallo-β-lactamases [10]. In vitro studies have shown that avibactam can regain the antimicrobial activity of ceftazidime against many ESBL, AmpC, KPC and OXA-48 producing *Enterobacterales* and MDR *P. aeruginosa* isolates [11,12]. CZA has been approved for complicated intra-abdominal infections, urinary tract infections, and hospital-acquired and ventilator-associated pneumonia caused by Gram-negative bacteria [13]. There is also evidence of successful use for the treatment of nosocomial Gram-negative infections with limited treatment options, including effectiveness against CRE and MDR *P. aeruginosa* [14]. Ceftolozane/tazobactam (C/T) combines a novel antipseudomonal cephalosporin with an established β-lactam β-lactamase inhibitor. C/T has shown activity against MDR *P. aeruginosa* and ESBL-E, but not against carbapenemase-producing organisms [10]. C/T has been approved for use against complicated intra-abdominal and urinary tract infections [13]. However, the most attractive use of C/T is perhaps the treatment of CRPA infections, also for off-label indications due to the lack of more active in-label alternatives [15,16], whereas in the future it may receive approval for hospital-acquired and ventilator-associated pneumonia [17].

Nevertheless, the emergence of resistance to both agents is increasingly recognized among CRE and CRPA [18] highlighting the need for accurate and widely accessible antimicrobial susceptibility testing (AST) to optimize treatment decisions. Recently, the Vitek 2 AST-XN10 card (bioMérieux, Marcy l’ Etoile, France), which includes CZA and C/T in its panel, has been launched, but evaluations on its performance are scarce [19,20]. Based on these grounds, we investigated the performance of the new Vitek 2 AST-XN10 as well as the gradient diffusion Etest compared to the reference broth microdilution (BMD) method for CZA and C/T AST of *Enterobacterales* and *P. aeruginosa* using the current EUCAST categorization breakpoints.

## 2. Results

The collection of test organisms was tailored to include bacteria that carry potent β-lactamases (ESBLs and/or carbapenemases) representing a variety of resistance phenotypes and cover a wide range of minimum inhibitory concentrations (MICs). Overall, 69 were challenge isolates carrying important resistance mechanisms and 31 were ESBL-negative and susceptible to carbapenems. In particular, they included 17 *P. aeruginosa* (11 CRPA), 57 *K. pneumoniae* (34 carbapenemase- and 10 ESBL-producing), 19 *Escherichia coli* (two carbapenemase- and nine ESBL-producing) and 7 *Enterobacter cloacae* (two carbapenemase- and one ESBL-producing) isolates (Table 1).

In total, 24% and 45% of strains, all of which belonging to the challenge isolates, were shown by BMD to be resistant to CZA and C/T, respectively, according to the EUCAST breakpoints. Namely, 5/17 (29%) *P. aeruginosa* isolates (all CRPA) were resistant to both CZA and C/T (MIC ≥16/4 and >32/4 mg/L, respectively). Concerning the *Enterobacterales* isolates, 19/83 (23%) were CZA-resistant (3/11 *K. pneumoniae* OXA-48, 1/11 *K. pneumoniae* KPC, 3/3 *K. pneumoniae* VIM, 4/4 *K. pneumoniae* KPC+VIM, 5/5 *K. pneumoniae* NDM, 1/2 *E. coli* OXA-48 2/2 carbapenemase-producing *E. cloacae*), while 40/83 (48%) were C/T-resistant (10/11 *K. pneumoniae* OXA-48, 11/11 *K. pneumoniae* KPC, 3/3 *K. pneumoniae* VIM, 4/4 *K. pneumoniae* KPC+VIM, 5/5 *K. pneumoniae* NDM, 4/10 *K. pneumoniae* ESBL, 1/2 *E. coli* OXA-48 and 2/2 carbapenemase-producing *E. cloacae*). Of note, the median C/T MIC was two two-fold dilutions lower than the modal CZA MIC (0.5/4 versus 2/4 mg/L) for *P. aeruginosa* isolates, while the modal C/T and CZA MICs of *Enterobacterales* isolates were identical (1/4 mg/L) (Table 2 and Table 3, Appendix A Appendix A).

Regarding Vitek 2, there was one *K. pneumoniae* KPC isolate for which the run was terminated without a C/T MIC value even after repeat testing and thus, it was excluded from further analysis for this particular agent (growth failure rate 1%). Similar to BMD, 23% and 44% of isolates tested were resistant to CZA and C/T, respectively. On the whole, the categorical agreement (CA) between Vitek 2 and BMD for CZA was 99% (99% for *Enterobacterales* and 100% for *P. aeruginosa*). No major errors (ME) were noted, whereas one (1%) very major error (VME) was recorded related to a *K. pneumoniae* NDM isolate (Vitek 2 and BMD MIC 8 and >16 mg/L, respectively). Accordingly, the overall CA for C/T was 98% (98% for *Enterobacterales* and 100% for *P. aeruginosa*). One (1%) ME (Vitek 2 and BMD MIC >16 and ≤0.25 mg/L, respectively) and one (1%) VME error (Vitek 2 and BMD MIC 1 and 4 mg/L, respectively) was observed involving two *K. pneumoniae* ESBL isolates. Concerning the *Enterobacterales* strains, both the median Vitek 2 CZA and C/T MICs were one log_2_ dilution lower than the corresponding BMD values (0.5/4 versus 1/4 mg/L, respectively). On the other hand, for *P. aeruginosa* isolates, both the median Vitek 2 CZA and C/T MICs were identical to the median BMD values (2/4 and 0.5/4 mg/L, respectively). The overall absolute agreement (isolates exhibiting identical MICs by both methods) was 74% for CZA and 65% for C/T. Consistently, the essential agreement (EA) rate was lower for C/T (96%) than for CZA (100%). Even when the analysis was performed per organism group, EA for *Enterobacterales* and *P. aeruginosa* was 100% and 100% for CZA and 91% and 100% for C/T, respectively. Based on the aforementioned, the AST-XN10 card of the Vitek 2 platform fulfills the ISO requirements to assess the in vitro susceptibility of *Enterobacterales* and *P. aeruginosa* to CZA and C/T using the current EUCAST breakpoints (Table 2 and Table 3).

The Etest gradient test yielded identical resistance rates with the Vitek 2 categorizing 23% and 44% of strains as resistant to CZA and C/T, respectively. The overall CA between Etest and BMD for both agents was 99% (CZA: 100% for *Enterobacterales* and 94% for *P. aeruginosa*, C/T: 99% for *Enterobacterales* and 100% for *P. aeruginosa*). No MEs were found. One (1%) VME was identified for CZA related with a CRPA (Etest and BMD MIC 8 and 16 mg/L, respectively) and C/T with a *K. pneumoniae* ESBL (Etest and BMD MIC 2 and 4 mg/L, respectively) isolate. For both the *Enterobacterales* and the *P. aeruginosa* strains, the median Etest CZA MIC was identical to the median BMD value (1/4 and 2/4 mg/L, respectively). On the contrary, the median Etest C/T MIC was one two-fold dilution higher (2/4 mg/L) than the corresponding BMD value for the *Enterobacterales* isolates and one two-fold dilution lower (1/4 mg/L) for the *P. aeruginosa* strains. The overall absolute agreement between methods was relatively low, namely 43% for CZA and 45% for C/T, while the EA rate was 93% for both antimicrobials. Nevertheless, when the analysis was performed per organism group, an EA of 85% and 6% VME rate was recorded for CZA and *P. aeruginosa*, whereas the EA for *Enterobacterales* and C/T was marginally beyond the ISO performance criteria (88%). Hence, the Etest strips might not fulfill the ISO performance criteria for AST devices of ≥90% EA as well as ≤3% for VME rate to assess the in vitro susceptibility of *P. aeruginosa* to CZA using the current EUCAST breakpoints (Table 2 and Table 3).

## 3. Discussion

The time lag between the approval of newer antimicrobial agents and their inclusion on commercial AST panels is considerable [23]. C/T was approved by the FDA in December 2014 and CZA in February 2015. Although AST was not an urgent priority based on the assumption that there would be minimal resistance to these new agents, CZA and C/T resistance and treatment failures have been described soon after their launch [18], questioning their empirical use without AST results. Commercial AST methods for CZA and C/T AST have relatively recently become available and systematic performance assessments have not yet been widely published. For instance, in 2020, the EUCAST C/T breakpoint for *Enterobacterales* has been revised, but gradient diffusion tests have not yet been evaluated using the new cut-off. In the present study, we evaluated the performance of two commercially available assays (Vitek 2 and Etest) versus BMD, for AST of CZA and C/T against challenging Gram-negative bacteria carrying important resistance mechanisms, using the current EUCAST breakpoints.

According to our findings, Vitek 2 overall performance for CZA AST of *Enterobacterales* and *P. aeruginosa* fulfilled the ISO performance criteria. Our results are consistent with a recent multicenter evaluation, revealing that the overall performance of Vitek 2 based on ISO criteria (applicable to EUCAST breakpoints) included CA, EA, ME and VME of 98.9%, 94.5%, 1.2% and 0%, respectively [20]. Of note, neither clinical nor challenge *Enterobacterales* isolates with the presence of OXA-like enzymes were tested in the latter study, as opposed to ours. On the contrary, an overestimation of CZA resistance in *P. aeruginosa* by the Vitek 2 system on the one hand and an unacceptable VME rate on the other have been recently reported (CA 83.5%, EA 89.0%, ME 18.1% and VME 8.8%) [19], which were likely attributed to the fact that 18.5% of the isolates tested had CZA MICs close (±1 log_2_ dilution) to the EUCAST breakpoint. Notably, ME and VME rates were calculated using the number of CZA-susceptible and -resistant strains, respectively, as the denominator. Nevertheless, differences in the dominator affect error rates and may give a false notion that higher error rates occur. Indeed, if the total number of isolates tested in the latter study is considered, MEs and VMEs decrease to 15.0% and 1.5%, respectively.

As far as C/T, to our knowledge, evaluation of Vitek 2 has not yet been performed for *Enterobacterales*. Based on our findings, Vitek 2 demonstrated CA, EA, ME and VME rates of 98%, 91%, 1% and 1%, respectively, thus meeting the ISO acceptance criteria to assess the in vitro susceptibility of *Enterobacterales* to C/T using the EUCAST breakpoints. Moreover, Vitek 2 appeared reliable to determine the in vitro susceptibility of *P. aeruginosa* to C/T when interpreted with the EUCAST criteria (CA 100%, EA 100%, ME and VME 0%), which is in line with the only available published data (CA, EA and ME of 95.5%, 96.5% and 1.2%, respectively) [19]. In contrast to our results for VME, the relatively high VME rate (12.5%) reported in the latter study could result from the enrichment of collection tested with strains exhibiting a resistance level around the EUCAST breakpoint (19.5% of isolates) and the fact that VME rate was assessed using the number of C/T-resistant strains as denominator. In fact, if the total number of isolates tested is considered, VMEs decrease to 2.5%.

MIC determination using gradient diffusion methods is very convenient for the clinical laboratories, however, only a limited number of studies have evaluated the performance of the Etest compared to BMD for CZA and C/T AST, particularly, using the EUCAST breakpoints. According to our findings, the CZA Etest for *Enterobacterales* isolates yielded a CA, EA, ME and VME of 100%, 97%, 0% and 0%, respectively, which is in agreement with those previously reported when using the EUCAST breakpoint (CA, EA, ME and VME of 100%, 100%, 0% and 0%, respectively) [24]. Εarlier observations have also supported the appropriate performance of the Etest for CZA AST against *Enterobacterales* following the CLSI criteria [25,26,27].

On the other hand, an EA of 85% and 6% VME rate were found for CZA Etest and *P. aeruginosa*. Additionally, applying the EUCAST breakpoints, previous studies have reported VME rates of 4.5–7.2%, corroborating our results [24,28]. Of note, no VMEs were noted with the CLSI breakpoints [25,26], highlighting the impact of the criteria applied for AST interpretation.

The present report is, to our knowledge, the first that evaluated the performance of Etest for C/T AST of *Enterobacterales* applying the 2020 revised EUCAST breakpoint. Excellent CA (99%), as well as ME (0%) and VME (1%) rates, were recorded while EA was somewhat lower (88%). Notably, earlier observations reported the Etest as a reproducible and accurate method for C/T susceptibility testing of *Enterobacterales* using the CLSI (R ≥ 8/4 mg/L) and the former EUCAST (R > 1/4 mg/L) categorization criteria [29]. Studies using larger numbers of clinical and challenge isolates are needed to assess whether the Etest is reliable for C/T MIC of *Enterobacterales* based on the current EUCAST breakpoint.

As far as the *P. aeruginosa* isolates, the C/T Etest yielded excellent results, which were similar to those previously found in a multicenter evaluation using the EUCAST breakpoint [29]. On the contrary, Daragon et al. reported higher VME (5.0% and 25.0% when the total number and the number of C/T-resistant isolates were used as the denominator for calculation, respectively) due to a high proportion of strains (19.5%) had C/T MICs close to the EUCAST breakpoint [19]. Following the CLSI criteria, previous studies have also confirmed the excellent performance of C/T Etest strips when applied to *P. aeruginosa* [27,29,30].

Our study has some limitations that need to be acknowledged; certain species were examined in small numbers and studied groups of strains were rather unequal. However, the strengths of our work include the use of standardized BMD panels as a reference method, as well as an adequate total number of Gram-negative isolates carrying challenge resistance mechanisms that were evaluated.

In conclusion, Vitek 2 was shown to be appropriate for AST of C/T and CZA against challenging Gram-negative bacteria. Additionally, the Etest performed quite well for C/T and *Enterobacterales*; it only exhibited marginally lower EA than the ISO criteria, which will probably not affect clinical decisions. On the contrary, unacceptable EA and VME rates were recorded for the CZA Etest and *P. aeruginosa* in our challenge collection, indicating that the respective susceptibility results should be interpreted cautiously until larger and geographically representative collections are tested. Clinicians may also consider CZA AST for CRPA by alternative methods, such as Vitek 2 or BMD, especially for patients at risk for CZA resistance. Continued monitoring of the performance of both methods is still recommended.

## 4. Materials and Methods

### 4.1. Bacterial Isolates

A total of 100 non-repetitive Gram-negative isolates, consisting of 83 *Enterobacterales* and 17 *P. aeruginosa*, were recovered from clinical samples of unique patients hospitalized in different wards of “Attikon” University hospital (Athens, Greece) in 2019, were tested. The clinical specimens included blood, bronchial aspirates, urine, deep tissue exudates and intra-abdominal secretions (Table 1). All isolates were recovered using standard-of-care culture media and were identified using the Vitek 2 system (Vitek 2 GN ID cards; bioMérieux, Marcy l’Etoile, France). Their susceptibility profile was also determined with the Vitek 2 system (Vitek 2 AST-GN cards; bioMérieux, Marcy l’Etoile, France) and the EUCAST guidelines were applied for the detection of resistance mechanisms and specific resistances of clinical and/or epidemiological importance [31]. All CRE were previously screened for carbapenemase production using: (i) meropenem 10 μg disks (Oxoid Ltd., Hampshire, UK) with or without inhibitors (phenylboronic acid, ethylenediaminetetraacetic acid) and (ii) the immunochromatographic assay NG-test CARBA 5 (NG Biotech, Guipry, France) [32]. ESBL producers were identified phenotypically by disk diffusion using cefotaxime and ceftazidime with and without clavulanic acid, according to the EUCAST protocols [31]. All isolates were stored at −70 °C in 20% glycerol storage medium (PanReac AppliChem, Darmstadt, Germany).

### 4.2. Susceptibility Testing

Prior to testing, the frozen isolates were sub-cultured twice on tryptic soy agar plates containing 5% sheep blood (Oxoid Ltd., Hampshire, UK), to ensure purity and viability. Each isolate was tested by the Vitek 2, Etest and BMD reference methods using a bacterial suspension that was prepared from a single fresh overnight culture using a digital densitometer (DensiCHEK; bioMérieux, Marcy l’Etoile, France). A suspension adjusted at 0.50–0.55 McFarland in 0.9% NaCl solution was used for the Etest and BMD protocols, while Vitek cards required a specific 0.45% saline solution (bioMérieux, Marcy l’Etoile, France). Inoculum density (colony count) and purity checks were performed on all isolates.

Vitek 2 was performed using the AST-XN10 card according to the manufacturer’s instructions. The concentration range tested was 0.25/4 to 32/4 mg/L and 0.12/4 to 16/4 mg/L for C/T and CZA, respectively. The gradient diffusion test was evaluated using the Etest strips on Mueller–Hinton agar plates (bioMérieux, Marcy l’ Etoile, France) in strict accordance with the manufacturer’s package insert. The ceftazidime and ceftolozane concentration gradients ranged from 0.016 to 256 mg/L with avibactam and tazobactam at a fixed concentration of 4 mg/L. Bacterial growth was inspected visually by two independent evaluators after 18 ± 2 h of incubation at 35 ± 2 °C in ambient air. The MIC result was read where the inhibition ellipse edge intersected the strip; if the ellipse intersected between two MIC values, a higher MIC was reported. Both methods were performed in singlet on the same day. The MICs of ceftazidime (concentrations tested 0.125 to 16 mg/L) and ceftolozane (concentrations tested 0.25 to 32 mg/L) with 4 mg/L avibactam and tazobactam, respectively, were determined by the BMD method using frozen customized antibiotic panels (Thermo Fisher Scientific, Cleveland, OH, USA). BMD was performed in triplicate using a single inoculum for each strain and results were evaluated by visual inspection by two blinded observers after incubation for 18 ± 2 h at 35 ± 2 °C. Discordance in both the Etest and BMD MICs was arbitrated by a third reader.

Susceptibility data were interpreted according to the current EUCAST clinical breakpoints for *Enterobacterales* and *P. aeruginosa* (CZA: *Enterobacterales* and *P. aeruginosa*, susceptible (S) ≤ 8/4 mg/L; resistant (R) > 8/4 mg/L, C/T: *Enterobacterales*, S ≤ 2/4 mg/L; R > 2/4 mg/L, and *P. aeruginosa*, S ≤ 4/4 mg/L; R > 4/4 mg/L) [21]. *E. coli* ATCC 25922 and *K. pneumoniae* ATCC 700603 were included as quality control strains in every series of experiments and were within the acceptable range for all tests throughout the study.

### 4.3. Data Analysis

The CZA and C/T susceptibility rates as well as the MIC range, MIC_50_ and MIC_90_ (the concentrations that inhibited 50% and 90%, respectively, of the isolates) were determined for each method. BMD was considered the reference method for data comparisons. Thus, the modal MIC was used as the reference BMD result and if no mode was obtained for a specific agent and isolate, BMD was repeated in triplicate, using the mode of all six results as the reference BMD result. The Etest MIC results were rounded up to the next serial two-fold dilution value for comparison with the BMD.

Comparative performance was assessed using EA, CA, MEs and VMEs rates according to standard definitions and acceptable performance criteria for AST devices as per the ISO standard 20776-2 (Appendix A) [22]. Observations of ME and VME were checked on new subcultures from frozen aliquots to confirm results. EA and CA were re-calculated after the replication. Calculations of EA, CA, ME and VME were obtained following the resolution of discrepant results after repeat testing. If an error was solved, the initial result was not included in the calculations and the calculations were made with the adjusted MICs; if an error persisted, the initial result was included in the calculations of EA and CA. Isolates that terminated due to failed growth after repeat testing with the Vitek 2 were excluded from the analysis.

## Figures and Tables

**Table 1 antibiotics-11-00865-t001:** Clinical specimens’ source and distribution of bacterial isolates.

Organism (No of Isolates)	Clinical Specimens’ Source (No of Isolates)
Challenge (*n* = 69)	
Carbapenem-resistant *P. aeruginosa* (*n* = 11)	Bronchial aspirates (*n* = 9) Urine (*n* = 2)
Carbapenemase-producing *K. pneumoniae* (*n* = 34) *K. pneumoniae* OXA-48 *K. pneumoniae* KPC *K. pneumoniae* VIM *K. pneumoniae* KPC+VIM *K. pneumoniae* NDM	Blood (*n* = 26) Bronchial aspirates (*n* = 2) Urine (*n* = 3) Intra-abdominal secretions (*n* = 1) Deep tissue exudates (*n* = 2)
ESBL-producing *K. pneumoniae* (*n* = 10)	Deep tissue exudates (*n* = 1) Urine (*n* = 5) Intra-abdominal secretions (*n* = 4)
Carbapenemase-producing *E. coli* (*n* = 2) *E. coli* OXA-48	Blood (*n* = 2)
ESBL-producing *E. coli* (*n* = 9)	Urine (*n* = 5) Blood (*n* = 2) Bronchial aspirates (*n* = 1) Intra-abdominal secretions (*n* = 1)
Carbapenemase-producing *E. cloacae* (*n* = 2) *E. cloacae* MBL *E. cloacae* KPC	Blood (*n* = 1) Intra-abdominal secretions *(n* = 1)
ESBL-producing *E. cloacae* (*n* = 1)	Intra-abdominal secretions (*n* = 1)
ESBL-negative, carbapenem-susceptible (*n* = 31)	
*P. aeruginosa* (*n* = 6)	Blood (*n* = 4) Bronchial aspirates (*n* = 2)
*K. pneumoniae* (*n* = 13)	Bronchial aspirates (*n* = 13)
*E. coli* (*n* = 8)	Urine (*n* = 8)
*E. cloacae* (*n* = 4)	Bronchial aspirates (*n* = 4)

**Table 2 antibiotics-11-00865-t002:** Comparison of testing methods for the determination of susceptibility of *Enterobacterales* and *P. aeruginosa* isolates to ceftazidime/avibactam according to the current EUCAST breakpoints [21].

Organism (No of Isolates)	Assay	Resistance Rate	MIC (mg/L)			Performance			
			Range	MIC_50_	MIC_90_	CA	EA	ME	VME
*Enterobacterales*	BMD	23%	≤0.125–>16	1	>16	_	_	_	_
(*n* = 83)	Vitek 2	22%	≤0.125–>8	0.5	>8	99%	100%	0%	1%
	Etest	23%	0.03–>256	1	>256	100%	97%	0%	0%
*P. aeruginosa*	BMD	29%	1–>16	2	>16	_	_	_	_
(*n* = 17)	Vitek 2	29%	1–>8	2	>8	100%	100%	0%	0%
	Etest	24%	1–256	2	128	94%	85%	0%	6%
Total	BMD	24%	_			_	_	_	_
(*n* = 100)	Vitek 2	23%	_			99%	100%	0%	1%
	Etest	23%	_			99%	93%	0%	1%

BMD: broth microdilution, CA: categorical agreement, EA: essential agreement, ME: major errors, VME: very major errors. Rates that do not fulfill the ISO requirements (CA and EA ≥90%, ME and VME ≤3%) [22] are indicated in bold face and are underlined.

**Table 3 antibiotics-11-00865-t003:** Comparison of testing methods for the determination of susceptibility of *Enterobacterales* and *P. aeruginosa* isolates to ceftolozane/tazobactam according to the current EUCAST breakpoints [21].

Organism (No of Isolates)	Assay	Resistance Rate	MIC (mg/L)			Performance			
			Range	MIC_50_	MIC_90_	CA	EA	ME	VME
*Enterobacterales*	BMD	48%	≤0.25–>32	1	>32	_	_	_	_
(*n* = 83)	Vitek 2 *	48%	≤0.25–>16	0.5	>16	98%	91%	1%	1%
	Etest	47%	0.06–>256	2	>256	99%	88%	0%	1%
*P. aeruginosa*	BMD	29%	0.5–>32	0.5	>32	_	_	_	_
(*n* = 17)	Vitek 2	29%	0.5–>16	0.5	>16	100%	100%	0%	0%
	Etest	29%	0.5–>256	1	>256	100%	100%	0%	0%
Total	BMD	45%	_			_	_	_	_
(*n* = 100)	Vitek 2	44%	_			98%	96%	1%	1%
	Etest	44%	_			99%	93%	0%	1%

* The run was terminated without MIC value for a *K. pneumoniae* isolate even after repeat testing and thus it was excluded from further analysis. BMD: broth microdilution, CA: categorical agreement, EA: essential agreement, ME: major errors, VME: very major errors. Rates that do not fulfill the ISO requirements (CA and EA ≥90%, ME and VME ≤3%) [22] are indicated in bold face and are underlined.

## Data Availability

The datasets analyzed during the current study are available from the corresponding author on reasonable request.

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
