# Peer review of "Comparative Evaluation of Vitek 2 and Etest versus Broth Microdilution for Ceftazidime/Avibactam and Ceftolozane/Tazobactam Susceptibility Testing of Enterobacterales and Pseudomonas aeruginosa"

_antibiotics, 2022, doi:10.3390/antibiotics11070865_

Round 1

Reviewer 1 Report

Well written
Result part -Text and numbers could be modified to present in diagram
Methods - EA, CA, MEs, and VMEs definitions could be explain more in supplement for reader who is not familiar to microbiology technique

Author Response

Please see the attachment, where we present responses to all reviewers.

Reviewer 2 Report

The work deals with an important topic, which is the determination of drug sensitivity in the case of bacteria characterized by multi-drug resistance. The topic is topical, regardless of the research group in a given country.

The manuscript topic is interesting, but the work has many bugs. First of all, it was not specified which strains where they came from; information was generally included (blood, urine, etc.). Perhaps it is worth preparing a graph that clearly shows the origin of the material; it is crucial for drug susceptibility testing (EUCAST). The studied groups of strains are unequal in terms of numbers, and I believe it is of great importance in such studies.

Moreover, no statistical test has been performed that would clearly define the usefulness of the tested tests. Some literature items are worth updating, for example, item 8 (2018) shows that our most common P. aerugionsa infections are nosocomial infections. The reader has the impression that research is not thought over; there is chaos in the discussion. The manuscript definitely needs editing.

Author Response

(The authors gave the same response as above.)

Reviewer 3 Report

The manuscript by Papadomanolaki et al. performed a comparative evaluation of Vitek2 and Etest relative to broth microdilution for ceftazidime/avibactam (CZA) and ceftalozane/tazobactam (CT) antibiotic susceptibility test of Enterobacterales and P. aeruginosa. The authors reported that Vitek2 could be used as an alternative for antibiotic susceptibility tests of CT against gram-negative bacteria. Overall, the study is straightforward and suggests a possible alternative method for a better antibiotic susceptibility test.

Minor comment:

  1. The authors should create sub-tables for different bacterial isolates mentioned in the result. As it is result section is hard to follow. 

Author Response

(The authors gave the same response as above.)

Reviewer 4 Report

The manuscript by Papadomanolaki et al describes Comparative evaluation of Vitek 2 and Etest versus broth microdilution for ceftazidime/avibactam(CZA) and ceftolozane/tazobactam(C/T) susceptibility testing of Enterobacterales and Pseudomonas aeruginosa. Briefly, the authors concluded that Vitek 2 measurements of CZA and C/T susceptibility correlated closely with the reference BMD and its performance for Enterobacterales and P. aeruginosa exceeded the ISO requirements, indicating that it can represent a suitable alternative to BMD for susceptibility testing of Enterobacterales and P. aeruginosa. But Etest did not fulfill the ISO performance criteria of EA and VME for CZA and P. aeruginosa.

Overall I did not find this article suitable for the Antibiotics journal. There was similar work published in the 2021 JCM journal with much larger clinical sample isolates compared to the current study. The authors used also Etest but they also emphasize that future work will be needed. 

Author Response

(The authors gave the same response as above.)

Round 2

Reviewer 4 Report

Thank you for your explanation.